# Qualitative Evaluation of Causes for Routine *Salmonella* Monitoring False-Positive Test Results in Dutch Poultry Breeding Flocks

**DOI:** 10.3390/microorganisms9112215

**Published:** 2021-10-25

**Authors:** Eduardo Costa, Armin Elbers, Miriam Koene, Andre Steentjes, Henk Wisselink, Peter Wijnen, Jose Gonzales

**Affiliations:** 1Department of Epidemiology, Bioinformatics and Animal Models, Wageningen Bioveterinary Research, Houtribweg 39, 8221 RA Lelystad, The Netherlands; armin.elbers@wur.nl (A.E.); jose.gonzales@wur.nl (J.G.); 2Department of Bacteriology, Host Pathogen Interaction & Diagnostics Development, Wageningen Bioveterinary Research, Houtribweg 39, 8221 RA Lelystad, The Netherlands; miriam.koene@wur.nl (M.K.); henk.wisselink@wur.nl (H.W.); 3Veterinary Centre Someren B.V., Slievenstraat 16, 5711 PK Someren, The Netherlands; a.steentjes@vc-someren.nl; 4Poultry Practice de Achterhoek, Spoorstraat 88, 7261 AG Ruurlo, The Netherlands; p.wijnen@ppda.nl

**Keywords:** *Salmonella* monitoring, poultry breeding flocks, retesting, false-positive, positive predicted value

## Abstract

The *Salmonella* monitoring program, as outlined in the EU Commission regulation 200/2010, asks for repeated sampling in order to ascertain progress in achievement of the EU target. According to Article 2.2.2.2.c of this regulation, the competent authority may decide to do a resample and retest when it has reasons to question the results of initial testing. In the Netherlands, the competent authorities have been resampling and retesting all initial positive samplings for several years because of doubts about false positive initial test results. An analysis of population data in the period 2015–2019 indicates that 48% of initial samplings at the farm were classified as false positive after resampling and retesting by the competent authorities. A qualitative analysis, assessing factors that could be associated with the occurrence of false positives, indicates that cross-contamination during the sampling process by the poultry farmer is probably the most likely source. Cross-contamination of samples during transport from the farm to the laboratory and/or cross-contamination at the laboratory are also considered possible sources. Given the slightly non-optimal system-specificity of the *Salmonella* monitoring program, there is good reason to make, or consider, standard resampling and retesting of initial positive results by the competent veterinary authorities possible within the EU.

## 1. Introduction

*Salmonella enterica* is a bacterial agent affecting food-producing animals and causing human salmonellosis, varying from invasive infections to most commonly a self-limiting diarrheal illness [1]. The usual route for *Salmonella* infection in humans is via food consumption being one of the most important foodborne disease agents in Europe [2,3]. In the Netherlands, Mughini–Gras & Van Pelt [4] estimated that around 11% of human salmonellosis cases were attributed to broilers or broiler products consumption. An effective strategy to mitigate the risk of salmonellosis is complex, involving all steps of production, including the identification of positive flocks in higher levels of the production pyramid, such as breeding flocks [5].

The Netherlands follows the EU Commission regulation 200/2010 for routinely sampling, testing and identifying *Salmonella*-infected flocks [6]. Samples collected on the farm are either pooled fresh feces, boot swabs and/or dust samples. In practice, almost 100% of samples collected in Dutch poultry breeding flocks are boot swabs. Farms are sampled on average every three weeks, and the sampling is predominantly performed by the farmers themselves, who submit the material for bacterial examination on *Salmonella* to a laboratory that is designated by the competent veterinary authority, the Netherlands Food and Consumer Product Safety Authority (NVWA), according to the demands described in the Regulation on Recognition and Designation for Veterinary Laboratories in the Netherlands [7]. AVINED, the Dutch association for poultry production, has made available informative material on how to execute the *Salmonella* sampling [6].

Breeding flocks testing positive for *S.* Enteritidis, Infantis, Hadar, Typhimurium, or Virchow must be slaughtered or destroyed so as to reduce, as much as possible, the risk of spreading *Salmonella*, and eggs must be destroyed or treated in a manner that guarantees the elimination of *Salmonella* in accordance with Community legislation on food hygiene [8]. This continuous effort during several years brought the prevalence below 1% [9], which is the target proposed by the EU Commission regulation 200/2010, of adult poultry breeding flocks of *Gallus gallus* [6].

According to Article 2.2.2.2.c of EU Commission regulation 200/2010, the competent authority may decide to do a resampling and retesting if it has reasons to question the results of initial testing (suspicions of false positive or false negative results). In the Netherlands, the competent authorities have been resampling and retesting all initial positive samplings, which are undertaken by the farmers, for several years because of doubts about the occurrence of false positive initial test results. Laboratory diagnosis of *Salmonella* is traditionally undertaken by bacteriological culture. The diagnostic specificity of this test is 100% [10], when performed according to ISO 6579-1 [11]. However, this only represents the specificity of the laboratory test alone and not that of the system as a whole. The routine *Salmonella* monitoring system is composed of a sequence of several steps: unpacking boot swabs from its package, putting the boot swabs on the footwear of the farmer, sampling by walking around with the boot swabs, packaging of the boot swabs after sampling, transporting of the material samples, unpacking samples at the laboratory, and performing the bacterial culture tests, which consist of various steps: pre-enrichment in non-selective liquid medium, enrichment in/on selective media, plating out and identification, and finally, confirmation of identity of suspected colonies [11].

If contamination happens in any of those steps between the sampling and the laboratory reporting, this will reduce the system-specificity to below 100%. Consequently, the positive predictive value of a *Salmonella* test result, i.e., the probability of a flock being positive given the test is positive, would be reduced [12]. Even a small loss of system-specificity may have an enormous effect on the diagnostic accuracy when the *Salmonella* prevalence is very low, as observed in the Netherlands [13], entailing an erroneous identification of a *Salmonella* negative flock as positive.

The objectives of our evaluation of the Dutch *Salmonella* monitoring program in poultry breeding flocks were: (a) to get insights into a population estimate of false positive diagnostic results during *Salmonella* monitoring of Dutch poultry breeding flocks; (b) to qualitatively assess possible causes of false positive *Salmonella* results in the current routine sample and laboratory test procedures; (c) to demonstrate the effect of several scenarios of loss of specificity on the post-test probability of infection of a breeding flock following a monitoring routine positive test. In addition, a questionnaire was set out among veterinary poultry specialists within 22 European countries to ascertain their perspective on the occurrence of false positive results in the EU *Salmonella* monitoring program.

## 2. Materials and Methods

### 2.1. Salmonella Test Data

We received data comprised of all *Salmonella* routine monitoring samplings at adult poultry breeding flocks in the Netherlands for the period 2015–2019. The data consisted of (1) flock category: Parent Stock (PS) or Grand Parent Stock (GPS); (2) laboratory identification number, indicating the private laboratory that processed the samples from the initial routine sampling by the farmers, veterinarians or technicians; (3) unique identification number of the poultry farm; (4) postal code of poultry farm; (5) type of sample (boot swab, or other); (6) poultry house identification from which sample originated; (7) birth date of the flock in the poultry house; (8) date of sampling; (9) date of reporting the diagnostic test result by private laboratory; (10) diagnostic test result (*Salmonella* detected/*Salmonella* not detected); (11) *Salmonella* serotype, if *Salmonella* was detected.

In addition, we received the results of resampling and retesting by the NVWA. For resampling and retesting, the NVWA collected new samples at the farm, using the same collection method as the poultry farmers (virtually 100% boot swabs). Resamples were collected on average one day (min: 0, max: 3) after they received a report on the routine monitoring positive test result, and on average 4–6 days after the routine monitoring sampling was performed, predominantly by the poultry farmer. The NVWA data we received consisted of (1) identification number of the poultry farm; (2) postal code of poultry farm; (3) type of sample; (4) date of initial routine monitoring sampling by poultry farmer; (5) date of re-sampling (new samples collected) by NVWA; (6) date of reporting the NVWA retest test result; (7) retest test result (detected/not detected); (8) *Salmonella* serotype. We limited our investigation to the detection of *Salmonella enterica* serotypes listed in the EU Commission Regulation 200/2010: *S*. Entertitis, Infantis, Hadar, Typhimurium and Virchow [6].

### 2.2. Qualitative Assessment of the Probability of False Results

For this evaluation, we assume that the possible false outcomes in a *Salmonella* test arise from factors occurring, or actions taken, between the sample collection and the processing at the laboratory, causing discrepancy between the routine monitoring and confirmatory *Salmonella* test results. The assessment of the likelihood of various factors, given discrepant *Salmonella* test results, was based on conversations with stakeholders, bacteriologists, laboratory personnel and a veterinary practice with a large number of poultry farmer clients and extensive experience of *Salmonella* sampling of poultry breeding flocks by poultry farmers.

We provide an overview of possible explanations for the observed discrepancy between routine *Salmonella* monitoring positive results and the results of resampling/retesting by the NVWA. We distinguish between two main hypotheses: 

(1)The routine monitoring positive result is incorrect:

This means that, on the basis of the test results, it is incorrect in concluding that the poultry breeding flock is infected with *Salmonella*. False positive routine tests could potentially arise from six factors/actions: (i) (Cross-)Contamination during sampling by the poultry farmer; (ii) (Cross-)Contamination during transport of sampled boot swabs; (iii) Contamination of samples in the laboratory before or during diagnostic testing; (iv) Test characteristics; (v) Contamination of the poultry house without infection of the chickens; and (vi) Vaccination.

(2)The negative retest (confirmatory testing by the NVWA) result is incorrect: 

This means that, on the basis of the retest result, it is incorrect in concluding that the breeding flock is free from *Salmonella*. False negative confirmatory tests could arise from six factors/actions: (i) lack of sensitivity of the sampling performed; (ii) lack of sensitivity of laboratory testing (i.e., *Salmonella* concentration around or below the detection limit of the test); (iii) inactivation of *Salmonella* during transport to the laboratory; (iv) intermittent *Salmonella* excretion; (v) treatment of poultry with antibiotics; and (vi) acidification of drinking water after initial positive sampling.

Each of the identified factors was qualitatively assessed. The probability of their association with the specific assessed hypothesis was expressed qualitatively based on a scale used by EFSA [14], which is an adaptation of OIE [15]. The assessment probability scale has six levels: very high, high, medium, low, very low, negligible.

### 2.3. Assessment of the Post-Test Probability of Infection of a Poultry Breeding Broiler Flock Following a Monitoring Routine Positive Test

To assess the positive predictive value (PPV) of a routine *Salmonella* monitoring test, four scenarios of failure (f) were created, accounting for one cross-contamination out of 100, 1000, 10,000, and 100,000 collected and processed samples. The specificity (Sp) is the complement of the failure (i.e., Sp = 1 − f). The sensitivity (Se) used is 99% [16], and the prevalences *p* = 0.02%, and *p* = 0.04%, correspond to the frequencies of monitoring and retest positive samples, respectively; both are based on the data provided by NWVA. The PPV is calculated according to Greiner & Gardner [12]: PPV = (Se × p)/[(Se × p) + (1 − Sp) × (1 − p)].

### 2.4. Questionnaire

A questionnaire about the potential for cross-contamination of samples was sent out to poultry veterinarians of the Poultry Veterinary Study Group of the European Union (www.pvsgeu.org (accessed on 3 September 2021)). Members of this group are selected expert poultry veterinarians from the following 22 European countries: Austria, Belgium, Bulgaria, Cyprus, Denmark, Finland, France, Germany, Great Britain, Greece, Hungary, Ireland, Italy, Latvia, Netherlands, Norway, Poland, Portugal, Romania, Spain, Sweden, and Switzerland. The following questions were asked: In your experience as a poultry practitioner: (Q1) do you think that it is possible to get a false positive *Salmonella* result from a farm because samples taken by the farmer have become contaminated through cross-contamination during sampling? (Q2) do you think that it is possible to get a false positive *Salmonella* result from a farm because samples taken by the farmer have become contaminated through cross-contamination during transport of the samples from the farm to the laboratory? (Q3) do you think that it is possible to get a false positive *Salmonella* result from a farm because samples have become contaminated through cross-contamination in the process of handling and isolation in the lab? (Q4) Do you think that, given the existence of false positive *Salmonella* results in practice, it is important that all initial positive *Salmonella* results from a farm are confirmed by resampling and retesting from the same barn and farm? For each answer by a responder, the likelihood of the chosen answer (yes or no) is asked on a scale from 0 (not sure) to 10 (very sure). The last question was: If you have answered questions Q1 to Q3 with “Yes”, what would be your ranking of the likelihood of the appearance of false positives? The details of the questionnaire are depicted in Appendix A.

## 3. Results

### 3.1. Salmonella Test Data Summary and Exploration

The number of poultry breeding flocks in the Netherlands decreased over the years 2015–2019 (Table 1). During this period a total of 99,433 samplings were performed out of which 44 (0.04%), from 25 different farms, were initially positive. These 44 routine monitoring positive samplings were retested by the NVWA and 21 of them (48%) retested negative; therefore, they were judged as false positives. Hence, the apparent overall false positive rate of the system is equal to 0.02% (Table 1). The proportion of routine monitoring positive samplings was significantly lower (Mantel-Haenszel Chi-square test (stratified by year): *p* < 0.05) in Grand Parent Stock (GPS) compared to in Parent Stock (PS). This is an indication of differences in the way biosecurity is applied at the breeding farms, because it can be assumed that biosecurity measures are more strict in GPS, compared to in PS.

From the routine monitoring positive samplings that were confirmed as positive in the retest by the NVWA, the serotypes of the *Salmonella* strains isolated in both methods were identical (data not shown). Four poultry breeding farms showed recurrent occurrence of routine monitoring positive samplings that repeatedly retested negative by the NVWA: at the same farm in the same flock, or in a different flock with a different birth date, and in different sampling years. This might be an indication of a systematic problem with respect to introducing contamination during the sampling process.

### 3.2. Qualitative Assessment of the Likelihood of Routine Monitoring Positive Result Is Incorrect

#### 3.2.1. (Cross-)Contamination during Sampling by the Poultry Farmer

Conversation with a knowledgeable veterinary practice indicated that in some cases there are doubts about taking the correct precautions during sampling by poultry farmers. Comparison of the protocol on how to correctly perform sampling with boot swabs (see Appendix A) provided by the poultry industry for poultry farmers with the one used by the NVWA, suggests that the protocol from the poultry industry could possibly be improved to prevent contamination with *Salmonella* from outside the poultry house during the sampling process. Differences between both protocols are for instance the use of disposable gloves (as is the instruction of the NVWA sampling protocol, see Appendix A) before unpacking the boot swabs from the package and putting the boot swabs on, versus hand washing as described in the protocol of the poultry industry. Another difference is the use of plastic disposable over-boots before putting on the boot swabs as is the instruction of the NVWA sampling protocol, (see Appendix A), but not in the protocol from the poultry industry. In summary, the sampling protocol and hygienic procedures followed by farmers may not be enough to avoid cross-contamination of the sample. Therefore, the probability assessment for this factor was rated high.

#### 3.2.2. (Cross-)Contamination during Transport of Sampled Boot Swabs

There were interview reports of incidences in which boot swab samples were delivered at the laboratory in non-sealed, simple plastic bags (non-designated packaging materials (UN 3373 safety seal bags)) and/or use of non-designated methods of transport (UN 3373 conditions). This could, occasionally, lead to contamination along the way from other poultry samples or other possible sources during transport. These reports led to the probability assessment for this factor to be rated medium.

#### 3.2.3. Contamination of Samples in the Laboratory before or during Diagnostic Testing

In principle, a laboratory can be regarded as a much better controlled environment compared to a poultry breeding flock, farm or transport vehicle. Moreover, laboratories that have been designated by the Dutch competent authorities for performing diagnosis for the national *Salmonella* monitoring program, have to demonstrate that they operate competently and generate valid results. This approval demands, amongst other protocols, an ISO 17025 accreditation for detection of *Salmonella* according to ISO 6579-1, serotyping of these strains according ISO 6579-3 [17]. Additionally, in the NVWA-designated laboratories, laboratories are reviewed and rated by the National Reference Laboratory on their proficiency test performances. Results from recent *Salmonella* proficiency testing show that participating laboratories occasionally report false-positive results. An analysis of results of proficiency tests performed in the period 2019–2020, both from NVWA-designated laboratories and laboratories in other countries (e.g., a combination of laboratories from Italy, Belgium, Germany, Spain, Portugal, UK, Eire, Greece, Cyprus, Malta, Serbia, Croatia, Turkey, Morocco, Canada, USA, South Africa, Botswana, Malaysia, India, Brazil, Thailand, Brunei, Singapore, Chile), shows that the proportion of false positive test results is, on average, 2.3% for the NVWA-designated laboratories and 2.7% for laboratories in other countries (Table 2 and Table 3). Such false positive test results may be caused by incorrect identification of other bacteria as *Salmonella*, or through cross-contamination in the lab. The culture method followed by serotyping has been shown to be an extremely specific test. However, serotyping is not routinely performed in every laboratory, and proficiency test results may be entered as *Salmonella* spp., potentially lowering the specificity of the method. Also, levels of *Salmonella* in proficiency samples may be relatively high compared to field samples, increasing the risk for cross-contamination compared to field samples. Nevertheless, the results suggest that a false positive result might also be generated during daily laboratory practice. The probability assessment for this factor as explanation for false positive results in the Dutch monitoring program was rated as medium. It should be taken into consideration that, even if cross-contamination is a rare event, if the number of samples tested are high (as in the case in *Salmonella* monitoring), this will inevitably lead to a number of false positive test reports.

Based on the number of false positive (fp) and the total number of negative samples tested (to) during proficiency testing in Table 2 (for the NVWA-designated laboratories) and Table 3 (from several countries), we used a beta distribution to describe the uncertainty around the estimate of probability of false positive (pp): pp~Beta(fp + 1; to-fp + 1) [18]. For the NVWA-designated laboratories, the mean probability of false positive testing during proficiency testing by a laboratory is 0.025 ranging from 0.011 to 0.044 (95% confidence interval). For other countries, it is 0.027 ranging from 0.019 to 0.037 (95% confidence interval). False positive probabilities higher than 0.05 and lower than 0.003 are rare in both the NVWA-designated laboratories and in other countries (Figure 1).

#### 3.2.4. Test Characteristics

The *Salmonella* culture method, performed according to ISO 6579-1 [11], is considered a test with a 100% specificity, i.e., when *Salmonella* is not isolated from a sample, it can be assumed that *Salmonella* was actually not present in the sample. According to EU regulation 2160/2003, it is allowed to use an alternative test method instead of culture [6], provided it has been validated in accordance with EN ISO 16140-2 (validation alternative methods) [19]. To our knowledge, there is at least one commercially available polymerase chain reaction (PCR) test that complies with this and which is also used by a number of NVWA approved laboratories. Since this PCR test cannot distinguish between *Salmonella* serovars, this PCR test is always followed by bacteriological examination of the samples and serotyping of the isolated *Salmonella* if the test result is positive. Based on this, the probability of a false positive test result is very unlikely.

At the laboratory, incidental errors such as sample changes, contamination during testing, administrative errors and/or other actions during a test can occur. Since the laboratories approved by the NVWA for participation in the *Salmonella* monitoring program meet a certain quality standard, it may be assumed that these errors occur at a minimum. The probability assessment for this factor was rated very low to negligible.

#### 3.2.5. Contamination of the Poultry House without Infection of the Chickens

This could, for example, be a contamination entering the poultry house, whereby the *Salmonella* is unable to colonize the animals, but where the *Salmonella* is demonstrable (for a short time). Based on expert opinion, the probability assessment for this factor was rated very low.

#### 3.2.6. Vaccination

For prevention of *Salmonella* in poultry, vaccination is considered a valuable additional measure by increasing the resistance of birds against infection, and to decrease shedding of *Salmonella*. In Europe, both live and inactivated vaccines are available. Of the available *Salmonella* live vaccines, Salmovac^®^ offers the possibility of differentiation from field strains by means of a PCR test (performed by GD Animal Health Service, Deventer, The Netherlands). In other vaccines, the vaccine strains are unable to grow on the selective media used, so a culture method here should not give rise to false positive test results. Sources from the field indicate that the majority of flocks of broiler breeders in the Netherlands are vaccinated during the rearing period, mostly with live vaccines and, sometimes, (also) with an inactivated vaccine. However, there are still some questions remaining about the influence of vaccination on the possibility of low or intermittent excretion of *Salmonella*, thereby, demonstrating a temporary positive flock [5,20]. Based on the available information, the probability assessment for this factor was rated low.

In Table 4, a summary is given of the probability assessment and the sources for the assessment, as well as identified limitations in the assessment concerning an incorrect test result during initial routine *Salmonella* monitoring.

### 3.3. Qualitative Assessment of the Negative Retest (Confirmatory Testing by the NVWA) Result Is Incorrect

#### 3.3.1. Lack of Sensitivity of the Sampling Performed

This would mean that the *Salmonella* bacteria present in the flock would not be picked up during the sampling of the NVWA. According to ISO 6579-1:201, for boot swab samples the limit of detection (LOD_50_) was determined to be 3.8 cfu/sample (3.2–4.4 95% confidence interval) [11]. The sensitivity (Se) of the boot swab method for detection of infection at the flock level is dependent on the within-flock prevalence [16]; when this within-flock prevalence is high (e.g., >50%), the Se is expected to be very high (99%), provided that the sampling protocol is properly followed [16]. However, if the within-flock prevalence is low, the flock Se and the repeatability of the diagnostic system may be compromised. This means that with a low within-flock prevalence, two repeated samples from the same flock (taking within a short period of time) could give contradictory results (flock negative = all five boot samples negative; positive = at least one boot sample positive). However, subsequent sampling of such a flock (when prevalence is likely to be higher) would suggestively confirm a positive test result. Retesting is carried out by the NVWA protocol with supposedly a limited risk of sample contamination. This is confirmed by the historic results from subsequent routine monitoring samplings in the months after the (negative) retest by the NVWA confirmed the negative status of flocks. The probability assessment for this factor was rated very low.

#### 3.3.2. Lack of Sensitivity of Laboratory Testing (i.e., *Salmonella* Concentration in Samples around or below the Detection Limit of the Test)

If the number of *Salmonella* bacteria in the sample is close to the detection limit of the test, stochastic probability processes will play a role and samples from the same “population” return conflicting results. If a flock is infected with *Salmonella*, assumably, sufficient *Salmonella* bacteria will be present in the sample to be detected in the laboratory [16]. The five boot swab samples should be taken from different places in the poultry house, with each pair of boot swabs representing 20% of the floor area of the house. In the routine *Salmonella* positive samplings that were retested by the NVWA and tested positive, in 70% of those cases, 5 out of 5 pairs of boot swabs tested positive, indicating that if infection is present, it will be easily detected. The probability assessment for this factor was rated negligible.

#### 3.3.3. Inactivation of Salmonella during Transport to the Laboratory

Samples of the NVWA are conditioned according to the NVWA protocol and transported according to fixed protocols. Compared to the protocol and procedures followed by the farmers, the procedures followed by the NVWA are very strict and likely to minimize risks of loss of viable *Salmonella* in sample material. Therefore, the probability assessment of this event to happen resulting in false negative results is rated very low.

#### 3.3.4. Intermittent Salmonella Excretion

This is a possible factor at the animal level, as it is known that *Salmonella* can be excreted intermittently. At the flock level, the more animals are infected, the more constant the excretion pattern will be over time. However, with a low excretion of *Salmonella* in a limited number of animals, the chance of detection may be lower (low flock system-sensitivity) and, therefore, be more variable over time. The lack of positive laboratory results in subsequent samplings after a negative retest (as mentioned under *Lack of sensitivity of the sampling performed*), and the very short time between routine sampling and retesting, in combination with the results mentioned above (generally a positive retest shows *Salmonella* in almost all sampled boot swabs), the probability assessment for this factor was rated very low to negligible.

#### 3.3.5. Treatment of Poultry with Antibiotics

If antibiotics are administered to a flock in the event of an initial positive *Salmonella* test result, the flock is declared infected without further sampling and testing. Theoretically, this suggests that an antibiotic treatment could be applied by the poultry farmer, without this being officially stated. During the resampling process by the NVWA (after initial positive test by poultry farmer), five random chickens per flock are selected and tested at the laboratory for antibiotic residues. Results were at all times negative. The probability assessment for this factor was rated negligible.

#### 3.3.6. Acidification of Drinking Water after Initial Positive Sampling

The bacteria that are already present in the house are probably not inactivated by it, but this may lead to a reduction in the excretion by infected chickens, or loss of viability of excreted *Salmonella* possibly resulting in a negative culture result. This means that the probability of detection by the acidification in the relatively short period between the two samples will probably only be influenced to a very limited extent. The probability assessment for this factor was rated negligible.

In Table 5, a summary is given of the probability assessment and the sources for the assessment, as well as identified limitations in the assessment concerning an incorrect retest result.

### 3.4. Assessment of the Post-Test Probability of Infection of a Breeding Broiler Flock Following a Monitoring Routine Positive Test

Four specificity loss scenarios were considered, which assumed that the false positive rate, compared to the observed apparent false positive rate of 0.02% (Table 2), was 5 to 50 times higher (0.1% (Sp = 99.9%) to 1% (Sp = 99%)) or 2 to 20 times lower (0.01% (Sp = 99.99%) to 0.001% (Sp = 99.999%)). The post-test probability of infection or positive predictive value of the routine *Salmonella* test tends to decrease sharply at low prevalence (Figure 2).

The predictive value of the initial positive test result (PPV) at a 0.04% prevalence of infected farms would range, therefore, between 3.8% and 97.5% for Sp values ranging from 99% to 99.999%, respectively (Table 6). Considering the prevalence observed in retesting samples (0.02%), the positive predictive values range between 1.9% and 95.2% (Table 6).

### 3.5. Questionnaire

A total of 97 poultry veterinarians from 22 European countries were sent the on-line questionnaire by e-mail and 65 poultry veterinarians from 21 European countries (Austria, Belgium, Bulgaria, Cyprus, Denmark, Finland, France, Germany, Great-Britain, Greece, Hungary, Ireland, Italy, Latvia, Netherlands, Poland, Portugal, Romania, Spain, Sweden, and Switzerland) responded to the targeted questions in the questionnaire (response rate: 67%). About 90% of the responding poultry veterinarians indicated that they believe it is possible that cross-contamination during sampling by the farmer can lead to a false positive *Salmonella* test result. That statement was made with an average certainty of 79% (Figure 3). Furthermore, 70% of the responders indicated that contamination might happen during the transport of samples from the farm to the laboratory, with an average certainty of 63%. 94% of the responders indicated that they consider cross-contamination at the laboratory possible, with an average certainty of 81%. Ranking the possible sources of cross-contamination, respondents ranked first, sampling at the farm, followed by cross-contamination at the laboratory and then cross-contamination during transport (Figure 4). A total of 92% of the responders indicated that given the existence of false positive *Salmonella* results in practice, it is important that all initial positive *Salmonella* results from poultry farms are confirmed by resampling and retesting; that statement is made with an average certainty of 96%.

## 4. Discussion

This qualitative assessment has shown that the specificity (Sp) of the Dutch *Salmonella* routine monitoring system as a whole (system-specificity) is not perfect (i.e., less than 100%). In this assessment, the most relevant factors likely to affect this Sp are contamination/cross-contamination of samples during the collection of samples at the farm, transport of samples from farm to the laboratory and/or inadequacies during the laboratory processing. On the other hand, false negative results from the confirmatory tests were assessed to have a very low to negligible probability of happening. A deeper evaluation of the discrepancies between positive results in the routine *Salmonella* monitoring, testing negative when retested by the NWVA would demand following, and documenting with evidence (e.g., photo, video) all processes during the samplings on hundreds of farms. Therefore, a qualitative assessment was a reasonable approach, which led to highlighting potential flaws in the sampling and testing process that would need to be addressed in order to improve the specificity of the *Salmonella* routine monitoring system.

*Salmonella* can be introduced into the primary production chain in several ways, which may or may not lead to contamination of the poultry present. Risk factors for the infection of a flock are well documented, e.g., people (manure or dust particles on clothing, under shoes, in the hair, hands of the farmer etc.), vermin (mice, but also cats, dogs and insects can be contaminated), contaminated feed and insufficiently cleaned and/or disinfected equipment and utensils, such as tools, egg trays, crates and containers [5]. Such sources of contamination could also lead to contamination of a sample at any point.

Detailed examination of the present protocol from the poultry industry for poultry farmers on how to perform routine sampling with boot swabs indicates that the sampling process, following this protocol, is potentially prone to introduction of contamination from outside the poultry house (including cross-contamination from poultry to poultry house by, e.g., inappropriate cleaning and disinfection of hands). In such a scenario, as well as in scenarios with other ways of contamination, the laboratory test result is correct. Culturing *Salmonella* bacteria, followed by serotyping, is considered extremely specific [10,11], whereby there is hardly any doubt that *Salmonella* has been detected in the tested sample.

Although *Salmonella* culture is considered a test with a 100% specificity, results from the proficiency testing indicate that errors at the laboratories may occur. At this point, it is impossible to determine the source of these errors but any error, contamination during testing, administrative errors and/or other unintended errors when performing a test can, influence the result. Reports from the National Reference Laboratory for *Salmonella* in the Netherlands on the performance of proficiency tests show that false positive test results in proficiency testing occasionally occur. In order to gain more insight into this, further research should take place on the basis of detailed laboratory data.

Our assessment indicated that cross-contamination during the sampling process by the poultry farmer is probably the most likely source for the occurrence of false positive test results. Cross-contamination of samples during transport from the farm to the laboratory and/or cross-contamination at the laboratory are also considered possible sources. Unfortunately, the qualitative assessment does not allow quantifying the contribution of each of these steps (farm-transport-laboratory) to the overall false positive rate. The interviewed veterinarians ranked the likelihood of sample contamination to be the highest at farm level, followed by contaminations or testing errors at the laboratory. Contaminations during transport were considered the least likely to happen. This ranking is in agreement with our qualitative assessment and taken together, these factors suggest that measures at the farm and laboratory levels should be prioritized to improve the system-specificity of the surveillance system.

In the meantime, in response to our evaluation, the Dutch poultry industry commissioned the present poultry industry routine *Salmonella* sampling protocol to be redesigned, taking into account the possible flaws mentioned in the evaluation. Very recently, the new protocol was made available to poultry farmers [21]. It is highly recommended that communication of the new protocol by the poultry industry to the farmers is detailed and explicit in wording and visualization, using flyers, demonstration workshops performed by experienced veterinarians, and possibly making a video available on the website of the poultry industry.

The combination of the slightly non-optimal system specificity (>99% and <100%) of the *Salmonella* monitoring program and the low prevalence of *Salmonella* in poultry breeding flocks in the Netherlands can result in a predictive post-test probability of infection lower than 80%, even when considering a specificity of 99.99%, i.e., a failure rate of 1/10,000 samples. This low positive predictive value of a positive result during routine sampling brings not only economic but also ethical concerns. Given the low probability that a farm is actually infected, it is unethical to cull an adult poultry breeding flock based on an initial monitoring positive test result.

Even if sampling techniques and laboratory procedures are significantly improved, a certain level of false outcomes is still to be expected, with a high impact. The number of positive outcomes will most probably decline due to the improvements resulting in a lower probability of false positive results in the remaining positive test outcomes. However, it is justifiable to perform verification testing by resampling and retesting at a national reference laboratory as suggested in the article 2.2.2.2.c of the EU Commission regulation 200/2010 [6]. This confirmatory sampling step was also endorsed by poultry veterinarians from 22 European countries who participated in our questionnaire.

In all, the following recommendations are made: -Standard resampling and retesting of initial positive test results by the competent authorities for confirmation should be the norm;-Drastically improve the routine sampling protocol for poultry farmers; communicate the new protocol by showing how to properly sample in a video-film that is available on the website of the poultry industry and with a clear flyer with photo material;-Strict supervision of private laboratories by competent authorities, with a clear protocol as to what actions should be taken when laboratories repeatedly produce false-positive test results in *Salmonella* proficiency testing.

## 5. Conclusions

*Salmonella* prevalence in poultry breeding flocks in the Netherlands is low, and the PPV of an initial positive test is also low, which, in line with the retest findings of the NVWA, justifies an official resampling and retesting by the competent authorities. Various possible explanations for obtaining false positive results during routine *Salmonella* samplings have been provided and their probable contribution toward obtaining a false result assessed. In addition, we received responses to a questionnaire on the subject of false positive *Salmonella* test results of poultry breeding flocks from experienced European poultry veterinary experts. Taken together, the results of our assessment and those of the questionnaire, lead us to conclude that the most likely cause of the considerably high occurrence of false positive samplings (48% of initial positive samplings) is cross-contamination during the sampling process by the poultry farmer. Cross-contamination of samples during transport from the farm to the laboratory and/or false positive testing in the laboratory is also considered possible.

## Figures and Tables

**Figure 1 microorganisms-09-02215-f001:**
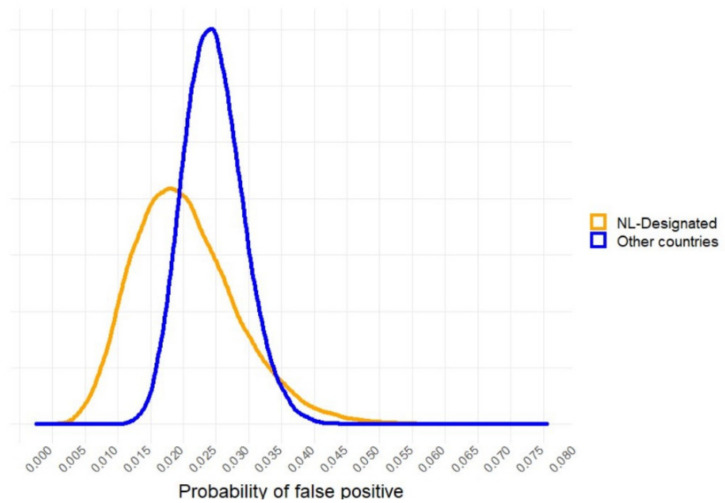
Uncertainty density distribution for the probability of false positive given a misclassification of a negative *Salmonella* sample at the laboratory, based on the results of the *Salmonella* proficiency tests in the period 2019–2020.

**Figure 2 microorganisms-09-02215-f002:**
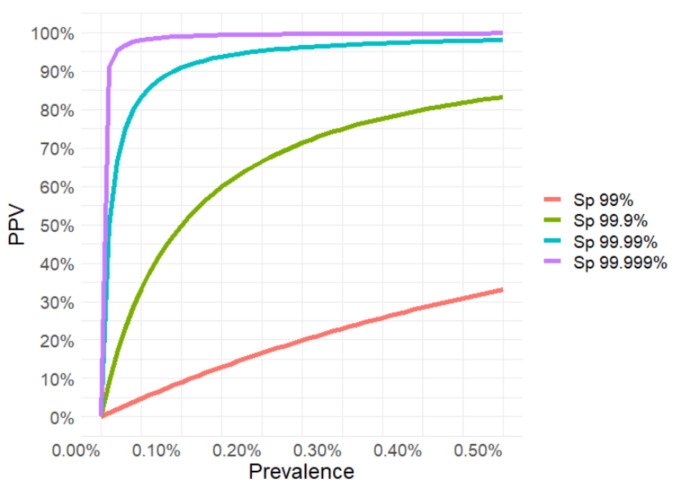
Relationship between the positive predictive value (PPV) of a positive test result and prevalence of a disease. For calculations, test specificities (Sp) from 99% to 99.999% and a test sensitivity (Se) of 99%, when the within-flock prevalence is higher than 10%-were used.

**Figure 3 microorganisms-09-02215-f003:**
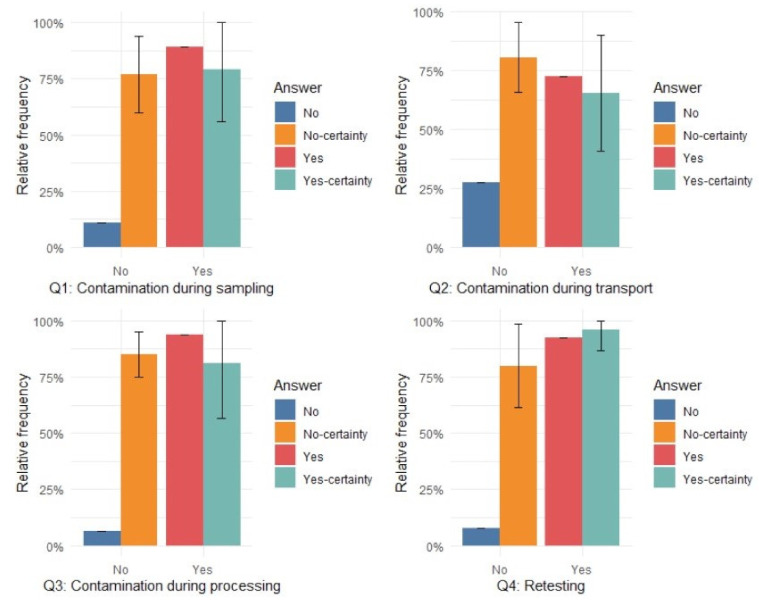
Poultry veterinarians (N = 65) responding to questions (Q1–Q3) on source of contamination of *Salmonella* samples in poultry breeding flocks leading to false positive routine *Salmonella* monitoring results and the need for making resampling and retesting possible of routine *Salmonella* monitoring positive results (Q4).

**Figure 4 microorganisms-09-02215-f004:**
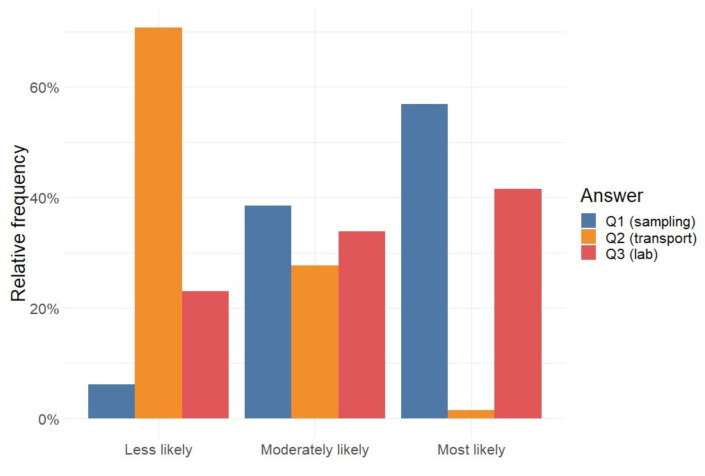
Poultry veterinarian (N = 65) opinions on the less, moderate, and most likely source of contamination of *Salmonella* samples in poultry breeding flocks leading to false positive routine *Salmonella* monitoring results.

**Table 1 microorganisms-09-02215-t001:** Results from the Dutch national control program for *Salmonella*; routine *Salmonella* monitoring positive samplings, NVWA-retest positive samplings (considered as true infections) and NVWA-retest negative samplings (false positive results) in relation to the total number (N) of *Salmonella* samplings in Dutch poultry breeding farms by year.

Year	Breeding Stock ^1^	N of Farms ^2^	N of Samplings ^3^	N of Routine Monitoring Positive Samplings	Serotype	N of Retest Positive Samplings	N of Retest Negative Samplings ^4^
2015	PS	347	13,175	14 (0.1%)	12 Enteritidis 2 Typhimurium	9 (0.07%)	5 (36% = 5/14)
2015	GPS	50	1533	1 (0.065%)	1 Enteritidis	1 (0.065%)	0
2016	PS	330	17,196	13 (0.08%)	13 Enteritidis	7 (0.04%)	6 (46% = 6/13)
2016	GPS	35	1145	0 (0%)		-	
2017	PS	321	20,465	1 (0.005%)	1 Typhimurium	0 (0%)	1 (100% = 1/1)
2017	GPS	35	2783	0 (0%)		-	
2018	PS	293	19,481	4 (0.02%)	1 Enteritidis 3 Infantis	0 (0%)	4 (100% = 4/4)
2018	GPS	31	3267	0 (0%)		-	
2019	PS	260	16,983	11 (0.06%)	6 Enteritidis 5 Infantis	6 (0.035%)	5 (45% = 5/11)
2019	GPS	29	3405	0 (0%)		-	
Total		1731	99,433	44 (0.04%)		23 (0.02%)	21 (48% = 21/44)

^1^ GPS: Grand Parent Stock; PS: Parent Stock. ^2^ A minority of farms had sometimes both a GPS and GP status within one year. ^3^ A *Salmonella* sampling is defined by a unique combination of poultry farm, sampling date, and poultry house within the farm that was sampled. Within a unique combination of sampling date and poultry house, one or more samples (with pooling of a maximum of five boot swabs into one sample submitted to the lab) were submitted to the laboratory. ^4^ Percentage of false positive samplings within initial positive samplings.

**Table 2 microorganisms-09-02215-t002:** Number (N) of *Salmonella* proficiency test results of NVWA-designated laboratories in the period 2019–2020, based on results from VETQAS proficiency tests (source: National Reference Laboratory for *Salmonella*, RIVM, Bilthoven, The Netherlands).

Year	2019				2020				
Quarter	1st	2nd	3rd	4th	1st	2nd	3rd	4th	Overall
N of negative samples in proficiency test	2	1	1	2	3	1	3	2	
N of positive samples in proficiency test	3	4	4	3	2	4	2	3	
N of laboratories	24	23	23	23	23	21	24	24	
N of false positives	2	0	1	2	1	0	1	1	8
Total number of negative samples tested (labs x negative samples)	48	23	23	46	69	21	72	48	350
% false positives	4.17	0.00	4.35	4.35	1.45	0.00	1.39	2.08	2.29

**Table 3 microorganisms-09-02215-t003:** Number (N) of *Salmonella* proficiency test results of laboratories from several countries in the period 2019–2020. (source: VETQAS, proficiency testing for veterinary laboratories; Animal and Plant Health Agency, UK.).

Year	2019				2020				
Quarter	1st	2nd	3rd	4th	1st	2nd	3rd	4th	Overall
N of negative samples in proficiency test	2	1	1	2	3	1	3	1	
N of positive samples in proficiency test	3	4	4	3	2	4	2	3	
N of laboratories	94	110	87	93	88	100	74	97	
N of false positives	2	3	3	9	1	2	11	3	34
Total number of negative samples tested (labs x negative samples)	188	110	87	186	264	100	222	97	1254
% false positives	1.06	2.73	3.45	4.84	0.38	2.00	4.95	3.09	2.71

**Table 4 microorganisms-09-02215-t004:** Summary of assessed factors/actions considered to be associated with an incorrect positive test result during routine *Salmonella* monitoring.

Factor/Action	Probability Assessment	Sources for Assessment and Limitations
(Cross-)Contamination during sampling by the poultry farmer	High	**Assessment**: Poultry industry sampling instruction compared with NVWA sampling protocol; interview with a veterinary practitioner with extensive experience in *Salmonella* sampling of his clients. Limitation: No documented follow-up of the procedures has been made so far.
(Cross-)Contamination during transport of sampled boot swabs	Medium	**Assessment**: Interview with a veterinary practitioner with extensive experience in *Salmonella* sampling of his clients. Limitation: No documented follow-up of the procedures has been made so far.
Contamination of samples in the laboratory before or during diagnostic testing	Medium	**Assessment**: ISO 17025 accreditation and results of proficiency testing by (VETQAS). Limitation: It lacks some specific evidence about the probability of contamination of samples at the lab.
Test characteristics	Very low to negligible	**Assessment**: Laboratories follow the internationally validated reference method ISO standard ISO 6579-1 for *Salmonella* isolation and identification. Limitations: No audits of laboratories are performed by NRL (RIVM); no overview of the diagnostic tests that are used and how these tests are deployed.
Contamination of the poultry house without infection of the chickens	Very low	**Assessment**: Expert opinion. Limitation: No specific evidence could be found and the assessment is based on expert opinion.
Vaccination	Low	**Assessment**: Expert opinion and literature evidence about vaccination. Limitation: The degree of *Salmonella* vaccination in the broiler breeding sector is unknown. Also, the effect on *Salmonella* shedding is still not clear.

**Table 5 microorganisms-09-02215-t005:** Summary of assessed factor/actions which could be associated with an incorrect retest negative test result.

Factor/Action	Probability Assessment	Sources for Assessment and Limitations
Lack of sensitivity of the sampling performed	Very low	**Assessment**: Based on [16], sensitivity of the sampling and testing is dependent on within-flock prevalence. If prevalence is higher than 10%, a high sensitivity of boot swabs method is expected. **Limitation**: No documented follow-up of the procedures has been made so far.
Lack of sensitivity of laboratory testing (i.e., *Salmonella* concentration in samples around or below the detection limit of the test)	Negligible	**Assessment**: Sensitivity of *Salmonella* may be hampered when the sample has a low concentration. It is expected that the concentration is high for positive flocks [16], which is supported by the NVWA retesting results of the routine *Salmonella* monitoring samplings testing positive. **Limitation**: No specific data to support the concentration of *Salmonella* in positive samples.
Inactivation of *Salmonella* during transport to the laboratory	Very low	**Assessment**: Samples collected by NVWA are conditioned and transported according to a strict NVWA protocol (presumed the golden standard description in Appendix A.). **Limitation**: No documented follow-up of the procedures has been made so far.
Intermittent *Salmonella* excretion	Very low to negligible	**Assessment**: There is a short time (maximally a few days) between routine sampling and retesting; with a low excretion of *Salmonella* in a limited number of chickens, the chance of detection may be lower. **Limitation**: More data about the length of excretion and interval between shedding cycles should be available.
Treatment of poultry with antibiotics	Negligible	Assessment: Five random chickens are included in every retest by the NVWA. (after initial positive test by poultry farmer) and tested for antibiotic residues; results were negative at all times. **Limitation**: Small limitation.
Acidification of drinking water after initial positive sampling	Negligible	**Assessment**: There is a very short time (maximally a few days) between routine sampling and retesting, limiting a hypothetical effect of acidification of drinking water. **Limitation**: Small limitation.

**Table 6 microorganisms-09-02215-t006:** Results of the four scenarios set to assess the positive predictive value (PPV) of the *Salmonella* routine testing in poultry breeding flocks in the Netherlands. The prevalences used are 0.02% and 0.04%.

Scenario (Failure Rate)	Sp	PPV (Prevalence 0.02%)	PPV (Prevalence 0.04%)
1–(1/100)	99%	1.9%	3.8%
2–(1/1000)	99.9%	16.5%	28.4%
3–(1/10,000)	99.99%	66.4%	79.8%
4–(1/100,000)	99.999%	95.2%	97.5%

## Data Availability

Data used for this study were made available by the poultry industry and the Netherlands’ Food and Consumer Product Safety Authority. Restrictions to the availability of these data apply due to privacy aspects of the poultry farms involved, the data were used under license and are not publicly available. Data are possibly available from the authors upon reasonable request and with permission of the poultry industry and the Netherlands’ Food and Consumer Product Safety Authority.

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
