# Peer review of "Qualitative Evaluation of Causes for Routine Salmonella Monitoring False-Positive Test Results in Dutch Poultry Breeding Flocks"

_microorganisms, 2021, doi:10.3390/microorganisms9112215_

Round 1

Reviewer 1 Report

The manuscript addresses the problem of arriving at false positive samples after testing for Salmonella enterica serovars. Samples positive  for Salmonella require a re-testing/re-confirmation by authorized laboratories. False reporting could have an impact on the poultry industry and could play a role for public health.  The authors provide a careful, sound and comprehensive assessment of the sampling procedures and processing of samples from poultry farms in order to test for presence of Salmonella in poultry flocks. They arrive at the main conclusion that the false positive outcome of the tests is large due to the behaviour and procedures on the poultry farms which could result in cross-contamination.

The paper is well written, the line of argumentation is clear and all conclusions are supported by the data provided. It would be nice if the authors could include (a) detailed suggestion(s) how to improve the current situation.

Minor point: Salmonella should be consequently be typed in italics.

Author Response

Dear reviewer. We addressed your questions. To make the process fast and easier we enumerated the comments:

  1. It would be nice if the authors could include (a) detailed suggestion(s) how to improve the current situation.

We appreciate the comment to include detailed suggestions to improve the current situation. The referee can find the new lines 580-590. Also, in lines 562-663 we included information about a new protocol released to farmers.

  1. Salmonella should be consequently be typed in italics.

We corrected all ‘salmonella’, changing it to ‘Salmonella’, including changes in references.

Reviewer 2 Report

The manuscript proposed by Costa and colleagues is very interesting and well-conducted. The paper highlighted the false-positive detection in the routinary surveillance against salmonellosis. 

I don't have any comments about the context of the manuscript but only a few things to fix:

  • table 1, 2,3: Please fix the headline. Avoid word wrap.
  • Appendix: Maybe is more useful convert them in Supplementary Materials

Author Response

Dear referee, we addressed your points enumerating the comments:

  1. table 1, 2, 3: Please fix the headline. Avoid word wrap.

We corrected all word separation in the headlines. In Tables 1, 2, and 3, we edited the title and the headings to shorten the headlines.

  1. Appendix: Maybe is more useful convert them in Supplementary Materials.

We appreciate this suggestion. All appendices are now supplementary materials.

Furthermore, we improved a few typos and wording structures in the text.
